# Variations of Sensorimotor Representation (Structure): The Functional Interplay between Object Features and Goal-Directed Grasping Actions

**DOI:** 10.3390/brainsci12070873

**Published:** 2022-06-30

**Authors:** Miguel Cienfuegos, Taeho Kim, Thomas Schack

**Affiliations:** 1Neurocognition and Action-Biomechanics—Research Group, Faculty of Psychology and Sports Science, Bielefeld University, Postfach 100131, 33501 Bielefeld, Germany; thomas.schack@uni-bielefeld.de; 2Center for Cognitive Interaction Technology (CITEC), Bielefeld University, 33619 Bielefeld, Germany; 3Department of Physical Education, Keimyung University, Daegu 42601, Korea; ksresearch77@gmail.com

**Keywords:** sensorimotor representation, object features, goal-directed actions, grasping, SDA-M

## Abstract

This study investigated the structure of sensorimotor representations during goal-directed grasping actions and explored their relationship with object features. Sixteen 3D-printed spheres that varied in size (i.e., a diameter of 20 mm, 40 mm, 60 mm, 80 mm) and weight (i.e., 40 g, 52 g, 76 g, 91 g) were used as experimental stimuli. The Structural Dimensional Analysis of Mental Representation (SDA-M) method was used to assess the sensorimotor representation structure during grasping. Participants were instructed in each trial to weigh, lift, or transport sets of two different spheres and to judge the similarity of the objects’ features, taking into account the executed grasping movement. Each participant performed a total of 240 trials, and object presentation was randomized. The results suggest that the functional interplay between object features and goal-directed actions accounts for the significant variations in the structure of sensorimotor representations after grasping. Specifically, the relevance of the perceived objects’ size and weight is closely interrelated to the grasping task demands and movement dynamics of the executed action. Our results suggest that distinct sensorimotor representations support individual grasping actions according to top-down influences modulated by motor intentions, functional task demands, and task-relevant object features.

## 1. Introduction

A fundamental aspect of human behavior that we effortlessly display in everyday life is the ability to grasp (i.e., achieve a specific hand position in gripping an object [1]) and manipulate objects within the environment. Gibson [2] documented how the intrinsic (e.g., weight, height) and extrinsic (e.g., size, location) features of an object influence behavior towards grasping. Several behavioral studies [3,4,5] have shown the influence of object features in the selection of the type of grip and the grasp kinematics. Other evidence suggests that intended goals produce different responses to the grasping behavior (e.g., grasp to hold vs. grasp to lift and place, grasp without intention vs. grasp to use) [3,6,7,8,9]. Much less is known about the relationship between object features and intentions [10], although a number of studies seem to suggest an established relationship between them [11,12,13,14]. From a theoretical point of view, the perceptual–cognitive (PC) approach [15] proposes that actions, including grasping, are planned and executed while guided by the formation of internal representations of perceptual information about these motor actions. Bernstein’s view reflects the general idea that movement control is based on representations and that these representations reflect the functional movement structure. Thus, the PC assumes that internal representations and motor actions are functionally related [16,17,18,19,20].

In the grasping domain, previous studies have shown that grasping movements are represented based on movement concepts and build on effect-oriented target codes (in relation to space rather than to the body) [21,22,23,24]. Typically, we require a representation of the perceptual patterns of the exteroceptive (e.g., object features) and proprioceptive and kinesthetics effects (e.g., sense of position and movement of the body) that result from the structure of the particular motion and refer back to the goal-directed grasping action. In other words, the PC hypothesizes the existence of a cognitive and a sensorimotor representation and its relevance for grasping.

Despite the importance of both the mental and sensorimotor representation systems as an information source for grasping and motor control, the perceptual–cognitive-based models [16,17,18] emphasize the role of hierarchically higher mental representations rather than sensorimotor representations. However, sensorimotor feedback has been found to help in weight discrimination [25,26,27], to modulate the grip size in prehension tasks [7], to produce a secure gripping in object manipulation [28], and to support decision-making in goal-directed movements [29]. Furthermore, the combination of sensorimotor feedback and visual information leads to faster and more precise grip apertures [30], supporting the idea that multisensory inputs are integrated during action planning and execution, resulting in optimized grasping movements. These results support the concept that sensorimotor feedback and the development of appropriate sensorimotor representations are pertinent for grasping movements and linked to motor performance.

Many everyday activities require that we grasp and manipulate objects (e.g., cooking, cleaning, and getting dressed). These grasping actions are associated with proprioceptive, kinaesthetic, tactile, optical, and auditory feedback (e.g., when cracking a nut). Although typically very little attention is paid to how we accomplish such movements, the inability to perform these actions (be it acquired [31], or congenital [32]) has dramatic consequences for our everyday lives [33]. Therefore, it is essential to understand how basic grasping movements are controlled and represented and how this can lead to the diagnosis of specific motor problems, and to develop a suitable intervention to maintain or rebuild a certain level of independence. Considering this last point, the need to thoroughly comprehend how sensorimotor representations are formed and structured on grasping actions may be raised.

A method that has proven effective in measuring representational structures is the Structural Dimensional Analysis—Motoric (SDA-M [34]). The SDA-M maps representations in a given set of basic action concepts (BACs) concerning goal-oriented actions, which are linked to functional and biomechanical components of motor actions. Importantly, this method allows us not to reveal representational structures by asking participants to give explicit statements but rather by using knowledge-based decisions in an experimental setting. The SDA-M approach has been applied in several different studies, such as manual actions [35], rehabilitation settings [36], cognitive robotics [37], and complex actions in sports contexts [38,39], to measure the structure and dimension of mental representation in motor actions. However, only a few studies examined the sensorimotor aspects of perceptual actions to understand how sensorimotor representations relate to grasping and how grasping movements are affected by object features [40]. So far, we do not know much about the role of grasping actions on the sensorimotor representation and the possible interplay between object features and goal-directed grasping actions.

The major purpose of this research was to study how sensorimotor aspects of perceptual actions (e.g., estimating the size and weight of objects) are related to object manipulation (e.g., reaching, lifting, transporting). We evaluated the role of goal-directed actions and the sensorimotor representation structures when performing manipulation tasks with natural objects. Based on the PC approach, we hypothesized that if sensorimotor-level representations play an important role as part of the cognitive reference structure underlying and guiding goal-intended motor actions. Tasks requiring different motor demands and triggering distinctive internal representations with the same set of objects will lead to different sensorimotor structures related to the specific motor output. We expected that the resulting sensorimotor representation structures would be influenced by the relevant object features and sensorimotor perceptions, according to task-specific demands. Additionally, we examined the influence and the relationship between object features and goal-intended grasping movements.

## 2. Materials and Methods

### 2.1. Participants

A total of twenty-seven students from Bielefeld University (mean age = 24.3 years, SD = 4.3, eighteen women, nine men) participated in this experiment. Participants were naive about the purpose of the experiment. All participants were right-handed (mean score = 96.7, SD = 14.9), as assessed using the Revised Edinburgh Handedness Inventory [41]. Participants gave their informed consent prior to the experiments, had normal or corrected to normal visual acuity, and did not have any known neuromuscular disorders. All participants received financial compensation for their participation. This study was conducted in accordance with local ethical guidelines and conformed to the 1964 Declaration of Helsinki.

### 2.2. Stimuli Design

The stimuli consisted of a set of three-dimensional (3D) printed spheres that were created for the experiment. The spheres’ 3D models were designed in the open-source parametric 3D modeller called FreeCAD. Each piece was prepared in two halves in the software MakerBot MakerWare^®^ and printed in a 3D printer, MakerBot Replicator 2X^®^, using ABS black filament. Subsequently, lead balls were fixed in the center of each half using wood glue to modify the weight of the spheres. Finally, the halves were joined together using plastic glue to form the spheres. The surfaces of the spheres were finely grained when necessary to avoid any external input—including thermal conductivity, texture, friction, and color—other than that haptically perceived [42]. The spheres differed systematically alongside two different dimensions (size and weight). The dimension differences used in this study were measurably greater than a discriminative threshold for size [5,43,44] and weight [25,45,46]. Each dimension comprised four levels as follows:Size (diameter): 20 mm, 40 mm, 60 mm, 80 mm.Weight: 40 g, 52 g, 76 g, 91 g.

The size of the objects was manipulated following arbitrary increments of 20 mm each, whereas their weight was increased following a logarithmic function (20% increments). These feature dimension levels were combined to create 16 objects (no object shared more than two dimension levels, i.e., there was no object with the same size and weight) and each feature dimension level appeared equally often (i.e., four times; see Table 1).

### 2.3. Measurement

A method that has been proposed to measure representation structures in long-term memory is the Structure Dimensional Analysis (SDA) method, which was originally developed by Lander and Lange [47] in cognitive psychology. The SDA method has been proposed for the psychometric analysis of the relational structures in a given set of concepts, and later adapted by Schack [21] for analyzing mental representations of movements (Structure Dimensional Analysis—Motorics, SDA-M). The method allows the revealing of representational structures not by asking participants to give explicit statements but rather by means of knowledge-based decisions in an experimental setting. Specifically, it is possible to investigate the clustering and relations of a set of objects regarding a specific motor action through the SDA-M. This experimental approach has been documented in several contributions [19,34,37,39,48]. In this study, we applied the SDA-M to measure sensorimotor representation structures.

The SDA-M consists of four steps. First is a splitting procedure to obtain distance scaling, a Euclidean distance, on the proximity between the representational object units, also called Basic Action Concepts (BACs) associated with a particular motor action in LTM. Second, a hierarchical cluster analysis transforms the given set of BACs into a hierarchical structure, a dendrogram. Third, a factor analysis linked to a specific cluster-oriented rotation process is performed to dimension the cluster solutions, resulting in a factor matrix classified by clusters. In the last step, an analysis of variance is run both between and within individuals’ differences to compare cluster solutions. For a more comprehensive description, refer to Schack [34].

For this study, sixteen BACs corresponding to the objects (described in the *Stimuli Design* section) linked with a particular motor action were used. In order to enumerate the objects, we combined the four size dimensions, S1, S2, S3, and S4, which have the diameters of 20 mm, 40 mm, 60 mm, and 80 mm, and the four weight dimensions, W1, W2, W3, and W4, with weights 40 g, 52 g, 76 g, and 91 g, respectively, to denote the objects in the form of S*i*W*i*, as shown in Table 2.

Following the SDA-M methodology, the first step - the splitting task - was performed on the representational distance between the selected BACs. The participants were asked to subjectively judge the functional equivalence of pairs of BACs (BAC × BAC: pairs of BACs are judged as “functionally related” or “not functionally related” to each other), verbally responding on a yes/no basis (i.e., yes if related, no if different) to the experimenter. *Functionally related* refers to the mobilization of body segments, muscles, and proprioception within an egocentric reference frame, according to their own motor memory execution of the movement. In other words, participants were required to judge whether two objects were related to one another or not regarding their grasping execution. Among the sixteen BACs chosen for this study, a randomly selected BAC was presented as an anchor; the remaining fifteen BACs were presented one after another in a randomized order. The same procedure was repeated, randomly selecting a different anchor BAC until every BAC was compared to every other BAC. In total, participants were asked to make a total of 240 judgments (16 anchors × 15 comparisons).

Subsequently, the method sums (i.e., the algebraic sum) the number of positive and/or negative decisions for each particular reference concept, forming subsets separately, and calculates the Euclidean distance scaling between the objects (BACs). The subsets are then transformed into *Z* values for standardization and finally combined into a *z*-matrix. The *z*-matrix forms the starting point for all further analyses.

In the second step, the *average-linkage* method is used to transform the *z*-matrix into an Euclidean distance matrix for hierarchical cluster analysis. This results in cluster solutions formed by the 16 BACs as a dendrogram. Each cluster solution is established by determining a critical Euclidean distance (d_*Crit*_). The d_*Crit*_ is used as a criterion to determine the significance level of each cluster. The significance level used in this study is 1% or <0.01. The d_*Crit*_ is represented as a horizontal line on the dendrogram. The cluster solutions that lie underneath the line are interpreted as statistically significant.

For the third step, the *z*-matrix is transformed into a correlation matrix, and then a factor analysis linked to a specific cluster-oriented rotation process is performed. This resulted in a factor matrix with the BACs (factors) and their weights (factor loadings) for the predominant cluster formation. Factor loadings can range from −1 to 1 and indicate how much a factor contributes to the given cluster. The incidental value for the factor charges C_*Crit*_ is defined by the correlations of the *z*-matrix and the number of factors [34]. The factor loadings above the C_*Crit*_ have the most relevance for the given cluster.

Finally, the fourth step of the SDA-M is to perform an invariant analysis using an invariant measure λ, and to perform a between-group comparison of the structural homogeneity cluster solutions to indicate whether differences are significant. The invariance measure is determined by the number of concepts in the cluster solution, pairwise cluster solutions, and the average number of clusters. The λ value can range from 0 to 1, with 1 indicating the most identical structure of the two cluster solutions. An alpha level of *p* = 0.05, as suggested by Lander [49,50], was used in this study; this means that the significant difference between two groups was set to λ < λ _*Crit*_ = 0.68 (for a more comprehensive description of the SDA-M methodology, please refer to Schack [34]).

### 2.4. Setup and Procedure

The present study consisted of three experimental conditions (described below): weight assessment, lifting, and transporting. Prior to the experiment, participants were informed about the two different feature dimensions in the spheres and were encouraged to explore them. The participants were randomly assigned to one of the groups, and performed a different task under each condition. All participants had five practice trials that preceded the start of the experiment. Participants sat comfortably in front of a table-top (height = 880 mm, depth = 760 mm, width = 1060 mm) for the duration of the experiment and performed the instructed task with the spheres using their right hand. The hand rested on a mark (120 mm diameter) on the right side of the table, with the hand fingers slightly opened. Spheres were positioned 300 mm from the front edge of the table at the participants’ midline, resting in a tee at the height of 80 mm for the weight assessment and lifting condition, and two tees at a distance of 400 mm were placed for the transporting condition (see Figure 1 for condition differences).

In the weight assessment condition, the given instruction was to assess the similarity of the executed task considering the weight of the objects and disregarding, as much as possible, their size (another feature dimension). To this end, participants had to memorize the perceived weight of each sphere when executing the task, without considering any other characteristic of the sphere (i.e., size). In summary, the task was not to discriminate the weight among the objects only but to consider the kinematics and sensorimotor perceptions of the task, in a similar fashion to Mccloskey, who asked participants to concentrate on their bodily sensations or the amount of effort needed [51]. Subjects were asked to reach out and pick up the object with their right hand as soon as the experimenter placed the sphere on the tee. Following this procedure, it was lifted to a height of approximately 40 mm and put back. The object was then replaced by a second sphere to go through the same process of being lifted and brought back to the initial position. Finally, each subject was asked to compare the second sphere to the first and respond verbally about whether they were “functionally related” or not to each other regarding the executed task, considering only the perceived weight. The experimenter recorded the answer on a pre-specified score sheet, for later use in the data analysis.

In the lifting condition, the objective was to memorize each sphere’s perceived weight and size and assess the similarity of the executed task. Similar to the weight assessment condition, subjects were instructed to execute the task as soon as the experimenter placed the sphere on the tee. Once the task was performed, a second sphere was brought to the working space to be lifted. Participants were asked to compare whether the sphere was “functionally related” to the other or not regarding their perceived weight and size for the performed lifting task.

In the transporting condition, the participants had to transport the spheres from their left side to their right side, memorizing the perceived dimensions (i.e., weight and size) of each sphere when executing the task and assessing the similarity of the movement by taking into account the weight and the size of the spheres. Subjects were instructed to reach out and grasp the object with their right hand as soon as the sphere was placed on the first tee by the experimenter. The subjects then transported the sphere to the second tee located 400 mm to the right. The first sphere was then removed, replaced with a second sphere, and the transporting task was repeated. After the task with the second sphere was finalized, participants had to judge whether the second sphere was “functionally related” or not concerning the sphere dimensions for the executed transport task.

For the three conditions described above, the participants were restricted to using only their right hand, and no other restriction was imposed (e.g. visual). Participants were free to execute the given task with their own selected form (e.g., grasping form, fingerprint landing, and position) and had unlimited time to complete the task. During the experiment, each sphere from the set was offered as an anchor (i.e., a reference object). Participants classified the remaining total *N*-1 spheres as functionally related or not functionally related regarding the executed task to the anchoring sphere. After all the judgements were made with this reference object, another object was presented as a reference object, and all other objects were compared to this reference object. Both the anchoring and the remaining objects for each decision, were arranged in a randomized order. Each condition comprised 16 blocks of 15 trials apiece, for a total of 240 trials (1 trial for each arrangement object) per participant. Trials in which the participant dropped one of the objects were immediately repeated.

### 2.5. Data Analysis

The structure of each grasping condition’s sensorimotor representation was determined by hierarchical cluster analysis (average linkage). Initially, the splitting procedure was performed (step 1 of the SDA-M). Based on the individual decisions obtained from the splitting procedure (yes/no choices), a distance scaling between each pair of BACs was established and added up to obtain an average inter-cluster distance that was then transformed into dendrograms outlining the structure of the BACs of the spheres. The cluster analysis aimed to reveal how many statistically significant clusters there were in the mean dendrogram by group condition and how well-structured their relations were [21]. For all cluster analyses conducted, the critical value d_*Crit*_ = 4.59 was chosen, which corresponds to a significance level of α = 0.01. Hence, all the clusters in the dendrograms below this value were considered statistically significant.

Then, the invariance analysis was performed to investigate general differences in the sensorimotor representation structure between conditions. Mean cluster solutions of each condition group were compared with every other group. However, this study did not pay attention to cluster solutions of individual participants within each group or inter-individual differences. According to Schack [34], two cluster solutions are significantly variant at λ < 0.68.

## 3. Results

### 3.1. Mental Representation Structure

The cluster analysis produced a mean group dendrogram for each tested condition. In detail, the mean group dendrogram of the “weight assessment” condition revealed three groupings of different objects (see Figure 2). The first cluster is formed with the objects belonging to the first two weight dimensions among all the size dimensions (BAC 1, 2, 5, 6, 9, 10, 13, 14). The second significant cluster integrates four objects of the third weight dimension among all the various sizes (BAC 3, 7, 11, 15). The third cluster includes three objects with the greatest weight and the first three sizes (BAC 4, 8, 12) among the set of objects. One object remained isolated, without being assigned to any of the clusters (BAC 16)—the sphere with the greatest weight and size.

The mean dendrogram of the “lifting” condition revealed four significant clusters (see Figure 3). The four clusters are formed with four elements each, combining two physical dimensions, the size and the weight. The first significant cluster is formed with the first two weight and size dimensions (BAC 1, 2, 5, and 6). The second significant cluster comprises the last two weight dimensions and the first two size dimensions (BAC 3, 4, 7, and 8). The third significant cluster is formed with the first two dimensions of the weight property and the last two dimensions of the size of the spheres (BAC 9, 10, 13, and 14). The fourth and final significant cluster integrates the “heaviest” weight and the “largest” size spheres of the set of objects (BAC 11, 12, 15, and 16).

The mean dendrogram of the “transporting” condition revealed three significant clusters (see Figure 4). The first significant cluster is formed with the first size dimension, including all the weight combinations of the objects (BAC 1, 2, 3, and 4). The second significant cluster includes the second size dimension with all the weight combinations of the objects (BAC 5, 6, 7, and 8). The third group is the largest grouping in this condition. It encompasses the last two size dimensions without considering the differences in the weight (BAC 9, 10, 11, 12, 13, 14, 15, and 16).

### 3.2. Invariance Analysis

Invariance analysis allowed us to determine whether there was a statistical difference between the condition groups. The invariance analysis indicated that there was evidently a significant difference between the “weighting” and the “lifting” conditions (λ = 0.47), between the “weighting” and the “transporting” conditions (λ = 0.46), and between the “lifting” and the “transporting” conditions (λ = 0.58). Table 3 shows a summary of the invariance analysis for the three conditions.

## 4. Discussion

The aims of the present study were twofold. First, we compared three different goal-directed grasping actions (e.g., weight assessment, lifting, and transporting) with a set of objects with feature variations (e.g., weight and size) to unravel the differences and similarities in the resulting sensorimotor representation structures. Second, we explored the interaction between goal-directed actions and object features in grasping. Our results revealed that the sensorimotor representation structures significantly differed among the goal-directed action conditions while performing with the same objects, supporting the established hypothesis. In addition, our findings showed a functional interplay between goal-intended actions and the object’s features when performing grasping movements.

For the weight assessment condition, we have found two unexpected outcomes. Firstly, the two lightest objects were aggregated together in one group. In other words, participants classified the action with these objects as functionally related, as if the weight difference was not meaningful for accomplishing the task. However, the execution with the heaviest and the second heaviest objects was functionally unrelated. The grouping of the lower weight dimension could be associated with the specific weight-related demands of the task with the lighter objects. When the weight demands increase, sensorimotor receptors (e.g., proprioceptive and kinaesthetic feedback) might become more relevant to the perception of limb position and movement dynamics, influencing the judgement of the participants [13,25,52]. In sum, the heavier the object, the more demanding the movement is, hence facilitating differencing proprioceptive and kinaesthetic perceptions during the action’s execution. Our second unexpected finding is that the action with the heaviest and largest object was not associated with any of the other objects. The literature suggests that size cue information may have a dominant effect in terms of how the weight of an object is perceived [53,54]. The larger size of the sphere may account for the way in which participants perceived the relationship of acting with this sphere compared to the rest. Our results indicate that higher weight demands facilitate weight discrimination between actions for a weight assessment task [55]. However, this is not to the extent of entirely neglecting the size–weight illusion effect (see [56] for a review), where size information is relevant for “feeling” how heavy an object is [54,57].

For the lifting condition, the results indicate a good balance between the size and the weight of the objects. The spheres with a small size and a small weight are grouped together. Similarly, the spheres with the largest size and greatest weight are clustered. Another group is formed by combining the two lightest and the two largest spheres, and the last group with the two heaviest and the two smallest spheres. Taken together, these results show an interaction involving the weight and size for a lifting task for building the sensorimotor representation. The findings are in line with those from Sartori and colleagues [13], where they suggest that the weight of objects interacts with their size during manipulation. Likewise, the lifting action seems to have an improved threshold to categorize the weight of the objects. The latter has been already suggested—Brodie et al. [25] proposed that afferent signals may account for this finer improvement during active lifting. Additionally, it has been suggested that the sensorimotor system may be more sophisticated, arising from the relative importance of each task [29,54]. Thus, the present findings during object lifting indicate that size information and weight perception are somehow integrated as a holistic object for judging the similarity of the lifting action. Although previous research suggests that the motor system related to lifting movement may run independently of the perceptual system for weight perception [26,27,58], our evidence points towards a form of linkage that enables an exchange of information between the two [59].

In the transport condition, the two smallest size dimensions were distinguished, while the largest two were combined together. The weight in this condition seems to have a minimal effect on how the actions were judged. At first glance, the result that the weight played nearly no role in transporting an object might be puzzling. One reason may be that the object’s size, shape, and, in general, the vision system provided sufficiently accurate and reliable information to complete the task correctly [30,60,61]. In our study, the spheres needed to be placed on a tee to finish the movement execution; the width of the tee and the size of the objects could have been the most relevant aspect to completing the task [62]. Consequently, the weight information may have been trivial for the movement completion, and the transporting was achieved relying primarily on visual information [30]. Another possibility is that the weight of the objects influenced some aspects of the tasks (e.g., the planning [13], the contact point position [63], and the applied initial forces [64]), but the weight information was less relevant for the most demanding part of the movement. Thus, the weight likely influenced the planning, gripping forces, and the initial phase of the transporting action, but its significance for the overall action effort and the end goal was not critical [9]. In conclusion, size and weight seem to play different roles during the movement execution. Our data show that the task demands, constraints, and end goal may modulate the perceived relevance of object features for judging the similarity of a transporting movement.

The role and relevance of size and shape in object control and manipulation have been extensively studied [7,8,60,65,66,67,68], as well as the mass and weight [9,13,63], and in goal-directed actions [11,12,69]. Our results expand previous works by suggesting that the relevance of the perceived objects’ size and weight is closely interrelated to the grasping task demands and movement dynamics of the executed action. In our view, it is natural to expect that the influence of a specific object feature differs according to the performed action. If so, it is likely that other influential factors related to the object and the action, e.g., material, texture, friction, movement dynamics, and configuration, might also be integrated during the sensorimotor perceptions according to the task demands and control of the action [64,70,71,72].

### 4.1. Study Limitations

A potential limitation in the present study was that we investigated only right-handed participants, using their dominant hand. We hypothesize that handedness might play a significant role in how the sensorimotor representation is structured. For instance, future studies may focus on comparing participants using both hands, or comparing groups with different skill levels for a specific grasping action (e.g., using an expert vs. novice paradigm [39]). Another factor that we expect to be relevant for further studies is to test contrasting visual deprivation conditions (e.g., blindfolded vs. uncovered eyes). Previous studies [73,74] have shown that visual deprivation may lead to enhanced performance in other sensory modalities, influence tactile acuity, and boost the plasticity of body representations. Other factors such as gender differences might also be explored. Finally, future studies should investigate other tasks (e.g., complex grasping actions) and other object features (e.g., shape, texture) than the one used in the present study (i.e., full body movement) to examine whether and how our findings generalize to different types of tasks and object features.

### 4.2. Implications for Research and Practice

Altogether, the presented findings provide compelling insights into the role of the visual, the sensorimotor system, and the sensorimotor representations for weight assessment, lifting, and transport grasping actions. Our data suggest that the functional interplay between object features and goal-directed grasping actions determined the significant variations in the sensorimotor representation structure. The present results are in line with the assumption that the control of voluntary action depends on the active representation of a goal-directed outcome [20,75,76]. Furthermore, our findings are consistent with previous studies of complex movement representations [37,38,48,77,78]. These studies emphasize that the internal representations and the processes in action control and motor execution are integrally interconnected. Hereof, our study is in line with the representational theoretical view on human action control during manipulation, as proposed in frameworks such as the predictive motor control [79,80,81], the ideomotor approach [82] (for a review of its history see [83]), the theory of event coding (TEC; [16,84]), or the cognitive action architecture approach (CAA-A; [21,85,86]). According to the latter model, the CAA-A proposes that movement control of goal-directed actions is planned and executed on the basis of the cognitive and sensorimotor representation levels and constructed hierarchically [21,85]. This approach views the functional construction of motor actions based on a reciprocal assignment of performance-oriented regulation levels and representational levels (see Figure 5). Therefore, our results may be interpreted within this framework, where internal representations and motor output are closely interlinked.

The present article provided insights into the interplay between object features and goal-directed actions by analyzing the difference in the sensorimotor representation structure in three grasping conditions. In doing so, we proposed a new experimental approach to capture the sensorimotor representation in grasping movements and the role of goal-intended actions and object features. We conclude by suggesting that distinct sensorimotor representations are involved in supporting grasping actions according to top-down influences (modulated by motor intentions, functional task demands or constraints, and task-relevant object features; [88,89,90]).

The importance of comprehensively understanding human grasping behavior is key to addressing how sensorimotor representations are related to motor control and how these underlying processes can be of use for developing suitable motor interventions in the rehabilitation domain, and for biomimetic robots and prosthetic tactile-related systems (e.g., systems inspired by human touch and proprioception)—for example, dexterous in-hand manipulation, virtual reality, rehabilitation [24,91,92]. A deeper understanding would allow for human-like sensorimotor control simulations in these systems, which could facilitate enhanced control of object manipulation and grasping behavior by integrating tactile and error feedback [91].

## 5. Conclusions

The results from our analysis indicate that task demands and end goals determine the variations in the sensorimotor representation structure and modulate how the similarity of the movement is perceived. These findings suggest a functional interplay between the goal-directed actions and object features and that sensorimotor representations support actions according to top-down influences (modulated by motor intentions, functional task demands, and task-relevant object features). Based on our results, we present a first step toward linking the cognitive mechanisms underlying goal-directed motor actions with the sensorimotor perceptual references of the unfolding grasping movement dynamics when manipulating objects. The implications of these findings are important for movement-related fields, including rehabilitation and physical therapy, tactile prosthetics, and biomimetic robots. While the present paper presents important steps toward linking sensorimotor representation with the goal-intend actions’ and object features’ interactions, much-needed work remains to extend the current findings to new task domains while also exploring the object features that modulate this relationship.

## Figures and Tables

**Figure 1 brainsci-12-00873-f001:**
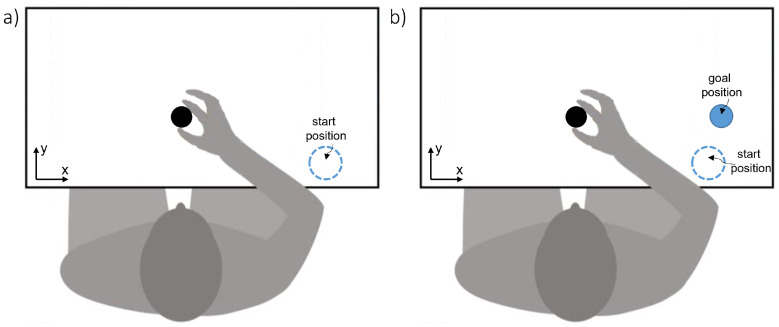
(**a**) Experimental setup for the “weight assessment” and “lifting” condition. Seated participants performed the instructed grasping movement with their right hand. Following the experimenter’s instruction, the sphere was taken from the starting position to execute the grasping action. Once completed, the sphere was released at its original position, and the participants returned to the start position. (**b**) Experimental setup for the “transporting” condition. Seated participants performed the instructed grasping movement with their right hand. Following the experimenter’s instruction, the sphere was taken from the starting position to execute the grasping action. The sphere was transported and released at the goal position, and the participants returned to the start position. (**a**,**b**) Once participants returned to the start position, a second object was brought by the experimenter, and the task was repeated. Participants were asked to judge the similarity of both objects’ features, taking into account the executed grasping movement. Finally, the experimenter recorded the answer on a pre-specified score sheet, terminating the trial. In all conditions, the participants had uncovered eyes and used their right hand only. The same sixteen 3D-printed objects were used throughout the experiment.

**Figure 2 brainsci-12-00873-f002:**
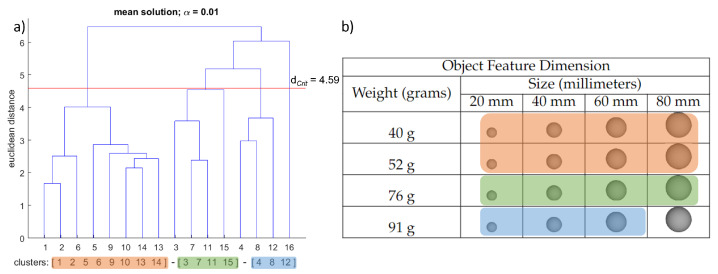
Results averaged among participants for the “weight condition”. (**a**) Mean dendrogram showing the resulting sensorimotor representation structure. The red horizontal line denotes the Euclidean critical distance. The critical value of the Euclidean distance was 4.59 at a 1% level of significance (i.e., d_*Crit*_ = 4.59, α = 0.01). Clusters that occur beneath the horizontal red line are statistically significant. (**b**) Spheres used in the experiment with their corresponding feature dimension levels. (**a**,**b**) Each color represents a significant cluster and its corresponding subset of spheres.

**Figure 3 brainsci-12-00873-f003:**
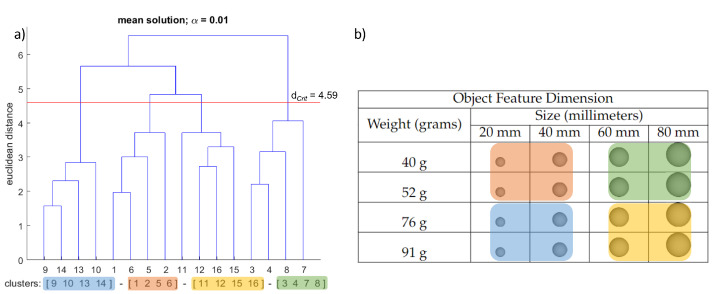
Results averaged among participants for the “lifting” condition. (**a**) Mean dendrogram showing the resulting sensorimotor representation structure. The red horizontal line denotes the Euclidean critical distance. The critical value of the Euclidean distance was 4.59 at a 1% level of significance (i.e., d_*Crit*_ = 4.59, α = 0.01). Clusters that occur beneath the horizontal red line are statistically significant. (**b**) Spheres used in the experiment with their corresponding feature dimension levels. (**a**,**b**) Each color represents a significant cluster and its corresponding subset of spheres.

**Figure 4 brainsci-12-00873-f004:**
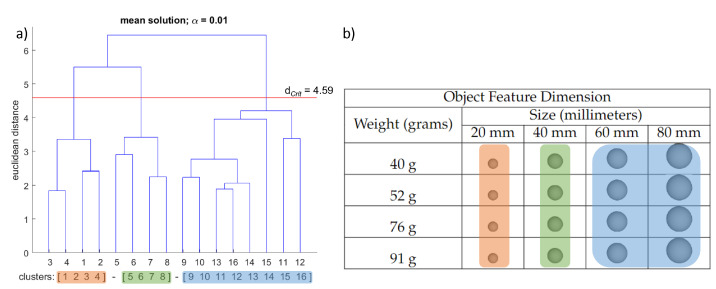
Results averaged among participants for the “transport” condition. (**a**) Mean dendrogram showing the resulting sensorimotor representation structure. The red horizontal line denotes the Euclidean critical distance. The critical value of the Euclidean distance was 4.59 at a 1% level of significance (i.e., d_*Crit*_ = 4.59, α = 0.01). Clusters that occur beneath the horizontal red line are statistically significant. (**b**) Spheres used in the experiment with their corresponding feature dimension levels. (**a**,**b**) Each color represents a significant cluster and its corresponding subset of spheres.

**Figure 5 brainsci-12-00873-f005:**
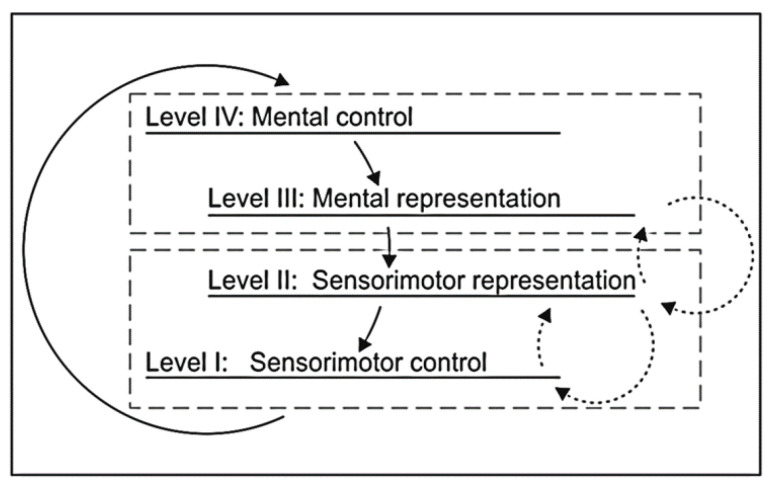
Levels of hierarchical action organization of motor actions. Adapted from Schack [21], reprinted from Kim et al. [87].

**Table 1 brainsci-12-00873-t001:** Set of sphere objects that differed in size and weight. Each sphere was 3D-printed in different sizes. Lead balls were fixed in the center of the spheres to vary their weight. The complete set of spheres was used in all conditions of this experiment.

Object Feature Dimension
**Weight (Grams)**	**Size (Millimeters)**
**20 mm**	**40 mm**	**60 mm**	**80 mm**
40 g	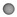	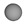	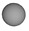	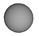
52 g	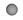	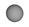	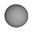	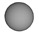
76 g	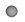	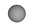	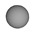	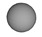
91 g	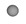	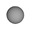	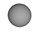	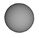

**Table 2 brainsci-12-00873-t002:** Basic action concept objects.

Number	BAC	Number	BAC
1	S1W1	9	S3W1
2	S1W2	10	S3W2
3	S1W3	11	S3W3
4	S1W4	12	S3W4
5	S2W1	13	S4W1
6	S2W2	14	S4W2
7	S2W3	15	S4W3
8	S2W4	16	S4W4

**Table 3 brainsci-12-00873-t003:** Invariance analysis results for the three conditions. The difference is significant for any value under λ = 0.68.

	Weighting	Lifting
**Lifting**	0.47	-
**Transporting**	0.46	0.58

## Data Availability

The data used for this study are available upon request to the corresponding author.

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
