# Peer review of "Variations of Sensorimotor Representation (Structure): The Functional Interplay between Object Features and Goal-Directed Grasping Actions"

_brainsci, 2022, doi:10.3390/brainsci12070873_

Round 1
Reviewer 1 Report
I was not able to understand the experimental procedure, what exactly were the instructions given to the participants, and what they had to answer (yes or no?). I also didn't understand the analysis of the results. I try to summarize what I understood. Participants were presented with 16 different balls, differing for size (20, 40, 60, and 80 mm) and weight (40, 52, 76, and 91 g). The balls were presented in pairs, one at a time. Participants were asked to perform one of two actions: "grasp and lift", and “grasp and move”. After “grasp and lift” participants were asked either to judge the similarity of both objects’ weight (i.e., respond verbally about whether they were "functionally close" … yes? No? on a Likert scale?), or to compare whether the spheres were "functionally close" regarding their perceived weight and size. After “grasp and move” participants were asked to judge the similarity of both objects’ dimensions (i.e., respond verbally about whether they were "functionally related") (is “dimensions” the size? Or both the weight and the size?). I don’t understand if "functionally close" means different than "functionally related".
I suggest to completely rewrite the Introduction and to explicitly state which is the aim of the experiment and how the results may contribute in terms of general knowledge of action programming. Furthermore, I strongly suggest to present and discuss the CAA only in the Discussion section, as a possible model to interpret results.
Furthermore, I suggest to explain analysis of data after having described the experimental procedure. In this way you can use some examples to explain better analysis of data.
As a general guideline, I suggest rewriting the article considering that readers may never have read an article by Thomas Schack, but may be interested in the influence that the relationship between weight and size of objects has on the comparison of similarity between objects, and in the role of different types of action execution in this comparison.
Reviewer 2 Report
Reading the manuscript written by Cienfuegos et al., was really interesting. The work aims to investigate the structure of sensorimotor representations during goal directed grasping actions and explored their relationship with object features. The results suggest that the functional interplay between object features and goal-directed actions accounts for the significant variations in the structure of sensorimotor representations after grasping. Specifically, the relevance of the perceived objects’ size and weight is closely interrelated to the grasping task demands and movement dynamics of the executed action. The authors’ results suggest that distinct sensorimotor representations support individual grasping actions according to top-down influences: modulated by motor intentions, functional task demands, and task-relevant object features.
Overall, this paper is written in professional English with sufficient introduction, detailed methods and solid data. However, typographical errors should be corrected (e.g lines: 91, 236, 243, 249, 350, 370, 374, 377, 389, 403, 303 etc). This will apply to the whole manuscript. Other typos include lack of space before/ after brackets.
Reviewer 3 Report
Dear Authors,
The manuscript titled "Variations of Sensorimotor Representation (Structure): The Functional Interplay Between Object Features and Goal-Directed Grasping Actions" is well written and logically constructed.
However, I have some question on the methods and experimental procedures.
1- First of all, I was wondering if the partecipants, during the experimental procedures, have their eyes covered or not. Indeed, especially for "transporting experiment" in which the task was to perceive the sphere dimension.
2- Did the Authors found differences between men and women? If not as I suppose, this should be added in the main text.
3- Did the Authors performed experiments with the same volunteer but letting them use the left hand? It should be interesting to know if there are differences.
4-Did the Authors performed experiments with left-hand volunteers (letting them use the left and the right hand as well)?
5-Finally, at page 6, line 204, the "z" of z-matrix should be written in italics as the Authors done in the other sentences.
Round 2
Reviewer 1 Report
The authors substantially modified the manuscript by improving it. However, they could have put more effort into explaining the basic theoretical principles to allow greater enjoyment even to non-expert readers. There are still many errors in the text (and also in the response to the referee), a sign of haste and little care.
Here are just a few examples of these errors:
Line 22: correct behavioural with behaviour
Line 103: BACs was not defined
Line 126: correct emphasizes with emphasize
Line 161-165: usually the past tense is used to describe the experiment. Other tense errors are present throughout the text (e.g., line 313, 319, …)
Lines 169-172 are redundant with lines 161-165: please integrate them
Line 177: I suggest to add: “We expect that the resulting sensorimotor…”
Line 243: Was the answer given orally? how was the response recorded? by pressing a button? with which fingers?
Author Response
We acknowledge and appreciate your comments. They have been taken into consideration for this article, and for the author's future research work.
Please see attached document.
